# Anomaly Detection in Multi-Agent Trajectories for Automated Driving

**Julian Wiederer**[1,2]   **Arij Bouazizi**[1,2]   **Marco Troina**[1]
**Ulrich Kressel**[1]   **Vasileios Belagiannis**[2]
[1]Mercedes-Benz AG   [2] Ulm University
{julian.wiederer, arij.bouazizi, marco.troina, ulrich.kressel}@daimler.com
vasileios.belagiannis@uni-ulm.de

**Abstract:**

Human drivers can recognise fast abnormal driving situations to avoid accidents. Similar to humans, automated vehicles are supposed to perform anomaly detection. In this work, we propose the spatio-temporal graph auto-encoder for learning normal driving behaviours. Our innovation is the ability to jointly learn multiple trajectories of a dynamic number of agents. To perform anomaly detection, we first estimate a density function of the learned trajectory feature representation and then detect anomalies in low-density regions. Due to the lack of multi-agent trajectory datasets for anomaly detection in automated driving, we introduce our dataset using a driving simulator for normal and abnormal manoeuvres. Our evaluations show that our approach learns the relation between different agents and delivers promising results compared to the related works. The code, simulation and the dataset are publicly available[1].

**Keywords:** Anomaly Detection, Multi-Agent Trajectory, Graph Neural Networks, Automated Driving

## 1   Introduction

A tired and inattentive driver often breaks the driving regulations by entering, for example, the opposite lane. This abnormal driving behaviour is usually early detected by the participating human drivers who react early enough to prevent harmful situations. Similar to humans, autonomous vehicles should perform anomaly detection as part of the automated driving modules [1, 2, 3]. Learning, thus, normal driving behaviour is necessary for detecting anomalies.

Anomaly detection is a long-developed approach in computer vision, for instance, to spot abnormal human behaviour [4, 5] or vehicle motion in traffic [6]. In robotics, the approach detects hardware failures of self-flying delivery drones [7] or helps a wheeled robot to navigate around unseen obstacles [8]. Although these approaches can be transferred to automated driving, they only consider a single agent in a static environment. That is barely the case for autonomous vehicles, where multiple agents influence each other through constant interactions. In this work, we present an approach to detect anomalies of multi-agents based on their trajectories.

We propose a spatio-temporal graph auto-encoder (STGAE) for trajectory embedding. Similar to the standard auto-encoder, it learns a latent representation of multi-agent trajectories. The main innovation of STGAE is the ability to simultaneously learn multiple trajectories for a dynamic number of agents. In a second step, we perform kernel density estimation (KDE) on the latent representation of the STGAE. We empirically observe that KDE captures well the distribution of the normal trajectory data. During the test phase, we detect anomalies in low-density regions of the estimated density. To evaluate our approach, we introduce a new dataset for multi-agent trajectory anomaly detection for automated driving. The current automotive datasets [9, 10] contain many hours of recordings, but lack anomalies due to the rareness of abnormal driving situations, whereas anomaly detection

---

[1]Project page: https://github.com/againerju/maad_highway

5th Conference on Robot Learning (CoRL 2021), London, UK.

datasets [11, 12] have the required anomaly labels but are not relevant to automotive and automated driving problems. Finally, scenario staging is often applied in behaviour modelling [13], but its prohibitive for driving anomalies since it will put the actors into life danger. For these reasons, we develop a multi-agent simulation and create a dataset with normal and abnormal manoeuvres. Then, we evaluate our method for single- and multi-agent configurations, including comparisons with deep sequential auto-encoder and linear models. Moreover, we rely on the standard metrics for anomaly detection to show that our approach delivers promising results compared to the related methods.

## 2 Related Work

**Multi-Agent Trajectory Modelling.** Trajectory prediction is essential for automated driving [14, 15]. Modelling the interaction with the environment and between the participants improves the prediction quality [13, 16]. The idea of information exchange across agents is actively studied in the literature [17, 18, 19]. For example, Alahi *et al.* introduced the social-pooling layer into LSTMs to incorporate interaction features between agents [20]. Recently, graph neural networks (GNN) have outperformed traditional sequential models on trajectory prediction benchmarks [21, 22]. GNNs explicitly model the agents as nodes and their connection as edges to represent the social interaction graph. Similarly, the social spatio-temporal graph convolution neural network (ST-GCNN) [5] extracts spatial and temporal dependencies between agents. Also, we use a related architecture to design our spatio-temporal graph auto-encoder for learning the normal data representation.

**Anomaly Detection.** On image and video data, anomaly detection is a long-standing topic. Handcrafted and motion features were traditionally employed for anomaly detection based on probabilistic PCA [23] or one-class SVM [24]. At the moment, deep neural networks dominate within the anomaly detection approaches [25, 26]. The usual approach is training an auto-encoder with normal data and then measuring the deviation of the test samples from the learned representation. For instance, Morais *et al.* [5] proposed a sequential auto-encoder to encode skeleton-based human motion and considered the reconstruction error as a measure for the detection of irregular motion patterns. Also, the variational auto-encoder (VAE) in [27] can learn the normal data distribution from a set of raw sensor signals in combination with a long short-term memory (LSTM) network. Unlike our work, these approaches assume a fixed number of input streams, i.e. sensor signals, instead of a varying number of trajectories, i.e. agents. Here, we formulate the idea of learning the normal data distribution with the spatio-temporal graph auto-encoder. Furthermore, we estimate the normal data density function instead of relying on the reconstruction error.

**Anomaly Detection in Trajectory Prediction and Control.** Anomaly detection has been extensively studied in robotics, e.g. model predictive control [28], collaborative robots [29], autonomous drones [7], robot navigation in crowds [30] and uncertain environments [31, 8]. Nevertheless, the prior work only address the problem of single agent anomaly detection. We tackle the problem of detecting anomalies in multi-agent trajectories.

## 3 Anomaly Detection in Multi-Agent Trajectories

We study the problem of anomaly detection for multi-agent trajectories. The input to our approach is a scene with $N$ agent trajectories of length $T$, where $N$ is dynamic over scenarios. We describe the observed trajectory of the agent $i$ with the agent states $\mathbf{s}^i = \{\mathbf{s}^i_t\}_{t=1}^T$, where $\mathbf{s}^i_t = (x^i_t, y^i_t)$ denotes the agent location in x- and y-coordinates. Our goal is to estimate the anomaly score $\alpha_t = [0, 1]$, i.e. normal or abnormal, for each time step of an unseen scene during testing, while only showing normal scenes during training.

We present a two-stage approach, where first a spatio-temporal graph convolution network auto-encoder is trained to represent normal trajectories in the feature space (Sec. 3.1). Second, we use the latent representation to fit a probabilistic density to the normal trajectories with kernel density estimation (Sec. 3.2). Finally, we present the anomaly detection score (Sec. 3.3) given the estimated density. The supplementary material provides visualisation of the training and inference process.

### 3.1 Spatio-Temporal Graph Auto-Encoder

We define the spatio-temporal graph auto-encoder (STGAE) as composition of the multi-agent trajectory encoder $g(\cdot)$ and the trajectory decoder $f(\cdot)$. The encoder maps a set of agent trajectories on the latent representation. The decoder transforms the latent representation back to trajectories.

**Trajectory Encoding.** The encoder is designed as a spatio-temporal graph convolution neural network (ST-GCNN) [32]. Given a set of $N$ agent trajectories of length $T$, we define the spatio-temporal graph $\mathcal{G} = \{\mathcal{G}_t\}_{t=1}^T$ as a set of directed spatial graphs $\mathcal{G}_t = (\mathcal{V}_t, \mathcal{E}_t)$. The spatial graphs model the multiple agents as nodes $\mathcal{V}_t$ and their connectivity as edges $\mathcal{E}_t$ to compute pairwise influence. The set of graph nodes $\mathcal{V}_t = \{\mathbf{v}_t^i\}_{i=1}^N$ represent the agent states in terms of the relative location $\mathbf{v}_t^i = (x_t^i - x_{t-1}^i, y_t^i - y_{t-1}^i)$. We define the edges $\mathcal{E}_t = \{e_t^{ij}\}_{i,j=1}^N$ to model how strong the node $i$ influences node $j$ at time step $t$. To this end, a kernel function $e_t^{ij} = \kappa_{edge}(\mathbf{v}_t^i, \mathbf{v}_t^j)$ [33] measures the similarity between two agents in the same time step defined as

$$e_t^{ij} = \begin{cases} 1/\|\mathbf{v}_t^i - \mathbf{v}_t^j\|_2 & , \|\mathbf{v}_t^i - \mathbf{v}_t^j\|_2 \neq 0 \\ 0 & , \text{otherwise.} \end{cases} \tag{1}$$

The influence is high for similar agent states and low otherwise. In the rare case of two agents sharing the same location, we set $e_t^{ij} = 0$.

We define the weighted adjacency matrix $\mathbf{A}_t \in \mathbb{R}^{N \times N}$ based on the connectivity parameters $e_t^{ij}$. We follow the procedure of Kipf *et al.* [34] and compute the normalised graph Laplacian $\hat{\mathbf{A}}_t = \mathbf{D}_t^{-\frac{1}{2}} \tilde{\mathbf{A}}_t \mathbf{D}_t^{-\frac{1}{2}}$ with the graph Laplacian $\tilde{\mathbf{A}}_t = \mathbf{A}_t + \mathbf{I}$, where $\mathbf{I}$ denotes the identity matrix, and the node degree matrix $\mathbf{D}_t$ with diagonal entries defined as $\mathbf{D}_t^{ii} = \sum_j \tilde{\mathbf{A}}_t^{ij}$. As introduced by Yan *et al.* [32], we aggregate over neighbouring agents using spatial graph convolutions $g_s(\mathcal{V}, \hat{\mathbf{A}}) = \sigma(\hat{\mathbf{A}} \mathcal{V} \boldsymbol{\theta}_s)$ with the activation function $\sigma(\cdot)$ and the network weights $\boldsymbol{\theta}_s$. We denote $\hat{\mathbf{A}}$ and $\mathcal{V}$ as the concatenation of the weighted adjacency matrices $\{\hat{\mathbf{A}}_t\}_{t=1}^T$ and the node features $\{\mathcal{V}_t\}_{t=1}^T$ over all time steps, respectively.

In dynamical systems, spatial features are not expressive enough since they ignore important temporal relationships. To include the time dimension, we connect the same node over consecutive frames using temporal convolutions $g_t(\cdot)$ as introduced in [35]. We define the encoder $g(\cdot)$ as a composite of spatial graph convolution and temporal convolution layers. As a result the encoder computes the latent representation $\mathbf{Z} \in \mathbb{R}^{T \times N \times F_g}$ with the latent feature dimension $F_g$.

**Trajectory Decoding.** Given the encoded representation, we define a decoder to reconstruct the set of input trajectories. The decoder applies multiple 2D convolution layers on the temporal dimension of the latent features [33]. To include agent interactions, the convolutions aggregate features across agents. We denote the decoder output as $\hat{\mathcal{V}} \in \mathbb{R}^{T \times N \times F_f}$ with output feature dimension $F_f$.

**Training.** For the training, we assume that the state of agent $i$ in time step $t$ comes from a bi-variate distribution, given by $\mathbf{s}_t^i \sim \mathcal{N}_2(\boldsymbol{\mu}_t^i, \boldsymbol{\sigma}_t^i, \boldsymbol{\rho}_t^i)$, where $\boldsymbol{\mu}_t^i, \boldsymbol{\sigma}_t^i \in \mathbb{R}^2$ are the mean and the standard deviation of the location, respectively, and $\boldsymbol{\rho}_t^i \in \mathbb{R}^2$ is the correlation factor. We estimate the parameters of the bi-variate distribution with the decoder output $\hat{\mathbf{v}}_t^i = \{\hat{\boldsymbol{\mu}}_t^i, \hat{\boldsymbol{\sigma}}_t^i, \hat{\boldsymbol{\rho}}_t^i\}$. We denote the estimated probability density function as $q(\mathbf{s}_t^i | \hat{\boldsymbol{\mu}}_t^i, \hat{\boldsymbol{\sigma}}_t^i, \hat{\boldsymbol{\rho}}_t^i)$ and train the model to minimise the negative log-likelihood

$$\mathcal{L} = -\sum_{t=1}^T \log\left(q\left(\mathbf{s}_t^i | \hat{\boldsymbol{\mu}}_t^i, \hat{\boldsymbol{\sigma}}_t^i, \hat{\boldsymbol{\rho}}_t^i\right)\right). \tag{2}$$

Similar to other sequential models [5], our STGAE requires inputs of a fixed temporal length. Therefore we clip each scene in the training set into smaller fixed-size segments of length $T'$ using a sliding window approach.

## 3.2 Kernel Density Estimation for Normal Trajectories

We rely on kernel density estimation (KDE) to approximate the probability density function of the normal trajectories from the latent feature representation of the STGAE. The idea is based on the assumption that normal trajectories fall in high density regions and anomalies occur in regions with lower density.

Given the trained STGAE, we encode the training segments and combine all latent representations in the set $Z_{kde}$. The KDE assumes all samples to be i.i.d. random variables drawn from an unknown true distribution $p$ [36]. We can approximate the true density of a new feature vector $\mathbf{z}$ by $\hat{p}$ defined as

$$\hat{p}(z) = \frac{1}{|Z_{kde}|h} \sum_{i=1}^{|Z_{kde}|} \kappa_{kde} \left( \frac{\mathbf{z} - \mathbf{z}_i}{h} \right) \text{ with the Gaussian Kernel [37] } \kappa_{kde}(\mathbf{x}) \propto exp \left( -\frac{\|\mathbf{x}\|^2}{2h^2} \right). \tag{3}$$

The kernel function $\kappa_{kde}(\mathbf{x})$ weights the observations differently based on the similarity to its neighbours. The parameter responsible for the weights of the points is the bandwidth $h$, which serves as smoothing. To sum-up, Eq. 3 computes the probability density of the agent's feature vector $\mathbf{z}$.

## 3.3 Abnormal Trajectory Detection

During the inference, we use the same sliding window approach as in the training and get the set of all test segments. For a test segment, the STGAE encoder computes the latent representation and the KDE from Eq. 3 estimates the density for each agent and time step, given the latent feature vector. We take the estimated density as a measure for anomalies. The feature decoder of the STGAE is not required during testing.

**Anomaly Score.** We follow a similar approach as introduced in [5] for the anomaly scoring. First, we compute $N$ anomaly scores $\alpha_t^i$ for all agents included in the same time step, and second compute the anomaly score $\alpha_t$ to measure if time step $t$ is abnormal. To score for one agent, we average the anomaly scores of all segments where the agent occurs by:

$$\alpha_t^i = \frac{\sum_{o \in S_o} p(\mathbf{z}_{t,o}^i)}{|S_o|}, \tag{4}$$

where $S_o$ are the overlapping sliding window segments in which the agent $i$ is present and $\mathbf{z}_{t,o}^i$ are the resulting feature vectors from the STGAE encoder at time step $t$. This results in one anomaly score for each agent in a specific time step. To identify if a time step is normal or abnormal, we compute the anomaly score $\alpha_t$ by taking the maximum over all agents:

$$\alpha_t = max(\alpha(\mathbf{z}_t^i))_{i \in 1,...,N}. \tag{5}$$

The max-operation avoids missing anomalies compared to the mean. We use $\alpha_t$ for the calculation of the metrics in our evaluations.

# 4 Experimental Results

We present the first dataset for anomaly detection in multi-agent trajectories and evaluate our method in comparison to seven baselines. See supplementary material for more results.

## 4.1 Dataset Development

To evaluate our algorithm, we propose the MAAD dataset, a dataset for multi-agent anomaly detection based on the OpenAI Gym *MultiCarRacing-v0* environment [38]. Since it was originally released as a multi-agent racing environment for learning visual control policies [39], we adapt it for anomaly simulation. We design the scenario of a two-lane highway shared by two vehicles, which

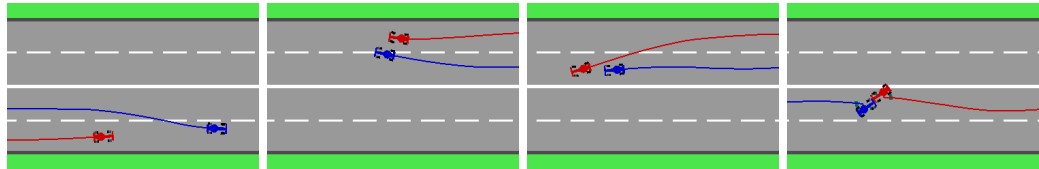

Figure 1: Example sequences from the proposed MAAD dataset. We show the observed trajectory of each agent. The first frame shows a save overtaking manoeuvre form the normal set. The other three frames contain abnormal actions: the blue vehicle is pushing the red vehicle aside, an aggressive reeving action of the red vehicle and a wrong-way driver who even collides with the upcoming traffic.

naturally leads to interaction, e.g. speed adjustments, lane changes or overtaking actions. The vehicles are controlled by human players to record multiple expert trajectories. For every sequence, we choose random initialisation of the agent starting positions to increase trajectory diversity.

In total, we create a dataset with 113 normal and 33 abnormal scenes. Beforehand, 11 different types of anomalies are defined in terms of breaking driving rules or careless behaviour[2]. Each abnormal scenario is recorded three times to incorporate more variations (see Fig. 1). After all recordings, the sequences are annotated with frame-wise labels by human experts with the ELAN annotation software [40]. We use 80 randomly selected normal sequences for training and the remaining 66 sequences to test. The sequences are sub-sampled to 10 Hz with a segment length of $T' = 15$ time steps, which corresponds to 1.5 seconds.

### 4.2 Baselines

Our baselines include *multi-agent* models as variants of our method, which explicitly model interaction, as well as *single-agent* models, ignoring interaction. The baselines can be further categorised as *one-class* and *reconstruction* methods.

**Single-Agent.** To examine the effect of interaction, we define four interaction-free models, two parameter-free and two neural network approaches. As simple parameter-free reconstruction method, we employ the constant velocity model (*CVM*) from [41]. Secondly, we approximate the trajectory with interpolation between the first and the last time step of the observed trajectory, we denote the model as linear temporal interpolation (*LTI*). The linear models succeed in the metrics, if the velocity profile of abnormal trajectories highly deviates from the normal trajectories. As a single-agent neural network, first we adapt the *Seq2Seq* model from [42]. It is composed of an encoder LSTM and a decoder LSTM. The encoder computes a feature vector representing one trajectory. After the last input is processed, the decoder tries to reconstruct the input trajectory from the feature vector. Next, we implement an interaction-free variant of STGAE by setting $A_t = I$. This reduces the model to a spatio-temporal auto-encoder, why we call it *STAE*.

**Multi-Agent.** Based on the proposed STGAE we evaluate three multi-agent baselines. For the first two variants, we use STGAE as reconstruction method. We train one STGAE with a bi-variate loss (*STGAE-biv*) and a second with classical MSE loss (*STGAE-mse*) on the trajectory reconstruction task. Similar to our approach, the third variant is a one-class classification method. We replace the KDE of our method with a one-class SVM (OC-SVM), which is an adaption of the traditional SVM to one-class classification. It takes the encoder features as input and finds a hyperplane separating the data points from the origin while ensuring that the hyperplane has maximum distance from the origin [43]. The baseline is denoted as *STGAE+OC-SVM*.

### 4.3 Evaluation Metrics

We quantitatively evaluate our approach following the standard evaluation metrics used in the anomaly detection literature [44, 45, 46], namely AUROC, AUPR-Success, AUPR-Error and FPR at 95% TPR. The AUROC metric integrates over the area under the Receiver Operating Characteristic

---

[2]We define the 11 anomaly sub-classes as *leave road*, *left spreading*, *aggressive overtaking*, *pushing aside*, *aggressive reeving*, *right spreading*, *skidding*, *staggering*, *tailating*, *thwarting* and *wrong-way-driving*.

Table 1: Comparison of the proposed method with the baselines on the proposed MAAD dataset using four metrics: AURC, AUPR-Abnormal, AUPR-Normal and FPR-95%-TPR. We differentiate between single- and multi-agent and further categorise into reconstruction and one-class classification methods. Highest scores are written in **bold**. For the trained models we provide the mean and standard deviation of ten runs. Note, CVM and LTI are deterministic with zero standard deviation.

| | Method | One-class vs. Reconstruction | AUROC ↑ | AUPR-Abnormal ↑ | AUPR-Normal ↑ | FPR-95%-TPR ↓ |
|---|---|---|---|---|---|---|
| Single-Agent | CVM [41] | reconstruction | 83.11 (±0.00) | 54.47 (±0.00) | 95.99 (±0.00) | 74.62 (±0.00) |
| | LTI | reconstruction | 75.47 (±0.00) | 48.89 (±0.00) | 92.37 (±0.00) | 95.03 (±0.00) |
| | Seq2Seq [42] | reconstruction | 56.15 (±0.68) | 16.92 (±1.01) | 89.54 (±0.13) | 84.62 (±0.26) |
| | STAE-biv $|A_t = I$ | reconstruction | 57.54 (±11.77) | 21.77 (±8.48) | 89.47 (±3.26) | 84.42 (±2.50) |
| Multi-Agent | STGAE-mse | reconstruction | 81.53 (±3.16) | 50.76 (±4.47) | 95.90 (±0.93) | 67.36 (±9.30) |
| | STGAE-biv | reconstruction | 74.82 (±5.10) | 37.79 (±7.16) | 94.10 (±1.31) | 77.80 (±9.77) |
| | STGAE-biv+OC-SVM | one-class | 85.97 (±2.40) | 52.37 (±8.85) | 97.11 (±0.59) | **49.90** (±6.33) |
| | Ours | one-class | **86.28** (±1.73) | **55.20** (±7.74) | **97.15** (±0.54) | 50.02 (±7.97) |

curve (ROC) and results in a threshold-independent evaluation. Note that a classifier with 50 % AUROC is equal to a random classifier, while 100 % is the upper limit and denotes the best possible classifier. We use the Area Under the Precision-Recall (AUPR) curve as our second metric. Other than AUROC, it is able to adjust for class imbalances, which is always the case in anomaly detection, i.e. the amount of abnormal samples is small compared to normal samples [45]. Here, we show both AUPR metrics, the AUPR-Abnormal, where we treat the abnormal class as positive, and the AUPR-Normal, where we treat the normal class as positive. Additionally we show FPR-95%-TPR, the False Positive Rate (FPR) at 95% True Positive Rate (TPR). For the one-class methods we directly apply the metrics on the output score and for the reconstruction methods we take the mean squared error (MSE) between the given and the reconstructed trajectory as anomaly score, following [5].

## 4.4  Implementation Details

The CVM approximates the agent trajectories assuming the same velocity for all time steps. The velocity is estimated given the first two time steps of a trajectory. For LTI the reconstruction error is defined as the distance between the ground-truth trajectory and equidistantly sampled locations on a straight line between the beginning and the end of the trajectory. Both, the encoder and decoder of the Seq2Seq model, are implemented with 3 stacked LSTM layers and 15 hidden features. We train the network for 1500 epochs using Adam optimiser with learning rate 0.01.

Our STGAE method is implemented as ST-GCNN encoder [33] and TCN decoder [35]. The encoder is composed of one spatial graph convolution layer and one TCN layer, both with five latent features, followed by the decoder consisting of five convolution layers for reconstruction. We train for 250 epochs using Stochastic Gradient Descent and learning rate 0.01, and decay the learning rate to 0.002 after 150 epochs. For evaluating STGAE-biv we sample 20 reconstructed trajectories from the bi-variate Gaussian distribution. STGAE with MSE loss does not require sampling.

Both one-class classification methods (OC-SVM and KDE) are implemented using a Gaussian kernel and the best hyperparameters are selected via grid search. Note, that OC-SVM has a minor supervised advantage, since validation of the model takes place on a holdout test set, i.e. 20% randomly selected from the test data. Hyperparameter tuning of $\gamma$ and $\nu$ for OC-SVM is performed via grid search with $\gamma \in \{2^{-10}, 2^{-9}, ..., 2^{-1}\}$ and $\nu \in \{0.01, 0.1\}$. The bandwidth of KDE is selected from $h \in \{2^{-4.5}, 2^{-4}, ..., 2^5\}$ via 5-fold cross-validation with the log-likelihood score as in [47].

## 4.5  Results

We present our results in comparison to the baselines. Afterwards, we evaluate different anomaly types and finally the performance stability over input sequence length and beyond pairs of agents.

**Comparison with the Baselines.** The comparison of our method with the baselines is presented in Table 1. Additionally, we show the ROC curves of all approaches in Figure 2. Our STGAE-biv+KDE outperforms both the linear and the deep methods in three out of four metrics, namely AUROC, AUPR-Abnormal and AUPR-Normal, and is on par with STGAE-biv+OC-SVM for FPR-95%-TPR. The second one-class classification approach STGAE-biv+OC-SVM reaches similar per-

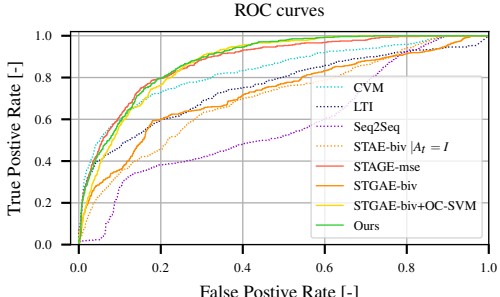
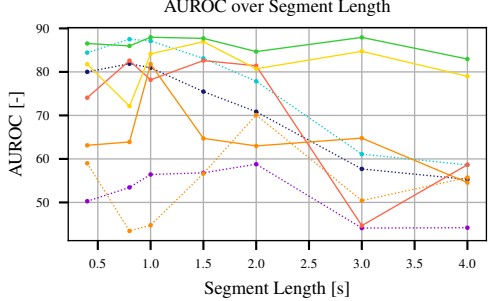

Figure 2: The ROC curves of the single-agent models (dashed lines) and the multi-agent models (solid lines).

Figure 3: AUROC metric over the segment length. Our method is more stable for various segment lengths.

Table 2: AUROC on different types of anomalies. We differentiate between non-interactive and interactive anomalies. The multi-agent models reach high-scores in both interaction and non-interaction anomalies. Highest scores are written in **bold**.

| Abnormal class | Non-interactive vs. Interactive | CVM | LTI | Seq2Seq | STAE-biv $A_t = I$ | STGAE -mse | STGAE -biv | STGAE-biv +SVM | Ours |
|---|---|---|---|---|---|---|---|---|---|
| leave road | non-interactive | 88.66 | 32.30 | 91.06 | 92.95 | 99.24 | 94.32 | **99.57** | 98.22 |
| left spreading | interactive | 90.88 | **96.90** | 47.06 | 55.06 | 91.92 | 52.76 | 94.16 | 96.56 |
| aggressive overtaking | interactive | 91.78 | 78.30 | 59.09 | 50.30 | 92.52 | 59.92 | 91.80 | **93.24** |
| pushing aside | interactive | 91.54 | 79.98 | 75.55 | 82.02 | **98.73** | 72.51 | 89.84 | 90.44 |
| aggressive reeving | interactive | **96.67** | 61.15 | 79.00 | 81.97 | 96.33 | 86.37 | 92.31 | 94.57 |
| right spreading | interactive | 87.66 | 89.83 | 45.22 | 61.75 | **96.33** | 53.57 | 95.55 | 96.24 |
| skidding | non-interactive | 96.90 | 94.69 | 90.00 | 98.84 | 98.31 | 70.24 | 98.80 | **99.65** |
| staggering | non-interactive | 86.87 | 85.30 | 57.83 | 72.64 | 90.58 | 66.53 | 94.91 | **96.01** |
| tailgating | interactive | 77.18 | 70.20 | 35.88 | 74.11 | **88.71** | 86.42 | 83.55 | 84.87 |
| thwarting | interactive | **98.08** | 90.75 | 88.34 | 92.56 | 92.68 | 96.58 | 81.95 | 81.59 |
| wrong-way driving | interactive | 63.16 | 64.63 | 68.86 | 53.90 | 61.46 | 57.88 | 73.11 | **73.15** |
| overall | | 83.11 | 75.47 | 58.06 | 70.01 | 87.70 | 72.67 | 87.38 | **88.34** |

formance and is the best for FPR-95%-TPR, but cannot reach the high-score in the remaining metrics. Considering the ROC curves, both one-class methods have a similar course with slight advantages for our method using KDE for anomaly detection. In general, the one-class methods are more stable over multiple runs. Our approach has practical advantage over the baselines as it remains superior even in consideration of the standard deviation of all other models.

Overall, besides STGAE-biv, the multi-agent models show higher accuracy compared to methods considering each agent individually. This indicates that some manoeuvres are only anomalous in the context of other traffic participants and could be considered as normal if the agent is alone on the street. This is also the reason for the performance drop for STGAE-biv, if the adjacency matrix $A_t$ is reduced to the identity matrix, i.e. no feature aggregation over neighbours. The linear models perform competitively, which gives rise to the difference between normal and abnormal lying in the degree of linearity of the trajectories. We empirically observe that Seq2Seq can not differentiate between normal and abnormal trajectories for most experiments, because it fails to learn the sequential dependencies. Overall, our approach can detect most of the anomalies compared to the baselines. In practice, high detection rates directly support the decision making process. Once an anomaly is detected the automated vehicle can react or pass the control to the human driver.

**Detection Score on Anomaly Types.** Table 2 shows the evaluation of the eleven anomaly types. To compute the ROC curve for an anomaly type, we consider this anomaly as positive and ignore all frames labelled with another anomaly category. Our method outperforms the others in four of the eleven classes and is competitive for the remaining ones. In particular, we significantly outperform the single-agent baselines on the *aggressive overtaking* category. We argue that it requires interaction modelling to distinguish the aggressive from a normal manoeuvre. Overall, the *wrong-way-driving* category leaves space for improvement. Including absolute coordinates in addition to velocity could help to learn more meaningful interaction features. Some abnormal trajectories can be detected without caring about interaction. This is when the linear models show their benefits, see *left spreading* or *thwarting* for LTI and CVM, respectively. The *left spreading* action is highly non-

Table 3: Comparison of STGAE-biv and our method on test sets with two, three, and four agents on the highway. Other than the reconstruction method, our one-class approach remains more stable in all metrics with increasing traffic density.

| Test Agents | Model | AUROC ↑ | AUPR-Abnormal ↑ | AUPR-Normal ↑ | FPR-95%-TPR ↓ |
|---|---|---|---|---|---|
| $N = 2$ | STGAE-biv | 69.08 | 39.28 | 90.07 | 92.19 |
| $N = 2$ | Ours | 92.34 | 66.75 | 98.48 | 35.20 |
| $N = 3$ | STGAE-biv | 78.20 | 42.52 | 94.73 | 78.23 |
| $N = 3$ | Ours | 91.26 | 63.57 | 98.36 | 35.38 |
| $N = 4$ | STGAE-biv | 52.42 | 17.95 | 88.03 | 88.19 |
| $N = 4$ | Ours | 89.41 | 60.52 | 97.87 | 42.55 |

linear, such that the LTI model fails to approximate correctly what results in high anomaly scores. Similar, *thwarting* results through strong braking, which is not following the constant velocity assumption of CVM. Again, STGAE-mse is the best reconstruction method and outperforms on three sub-classes, however with the downside of a computational expensive trajectory decoder.

**Ablation Study on Observed Sequence Length.** Figure 3 shows the influence of different segment length $T' \in \{4, 8, 10, 15, 20, 30, 40\}$ on the recognition performance. For the comparison we re-train all models on different segment length. Although we see a correlation between STGAE-biv and our method, our results remain stable for a large interval. As reported before, we reach the best performance for $T' = 15$. The performance of the linear models decreases for higher input length, which means that linear models are good trajectory approximators only for short sequences. In general, the reconstruction methods drop in performance for longer sequences. Both, CVM and LTI have peaks at 0.8 seconds, however our method remains the best overall with 88.34 % AUROC.

**Ablation Study on Scalability Beyond Pairs of Agents.** The proposed MAAD dataset includes diverse interactive anomalies between pairs of agents. However, the proposed model is flexible to the number of agents in a scene and can process more than two agents without adaption. We evaluate the recognition performance for a highway with higher traffic density on two models. In detail, we compare Our approach with the performance of STGAE-biv. Both models are trained on the original MAAD training set with two agents in each scene. For testing we create three ablation test sets with $N = 2$, $N = 3$ and $N = 4$ agents, see details in the supplementary material. The results are shown in Table 3. As before, our approach reaches higher scores compared to the reconstruction method. Interestingly, the metrics of the reconstruction method vary intensively with the number of agents. This indicates that once trained on two agents the reconstruction is confused by the features from the additional agents. It looks different for our method. Even though we see a small performance decrease with an increasing number of agents, our method is more stable and can reliably detect anomalies even on highways with higher traffic density.

## 5 Conclusion

We presented the spatio-temporal graph auto-encoder for trajectory representation learning. Our main contribution is the ability to simultaneously learn multiple trajectories for a dynamic number of agents. Our model learns normal driving behaviour for performing afterwards anomaly detection. To this end, we performed kernel density estimation on the latent representation of the model. During testing, we detect anomalies in low-density regions of the estimated density. Due to the lack of datasets for multi-agent trajectory anomaly detection for automated driving, we presented a synthetic multi-agent dataset with normal and abnormal manoeuvres. In our evaluations, we compared our approach with several baselines to show superior performance. Although our study is on driving trajectories, our approach can learn joint feature spaces in other multi-agent domains like verbal and non-verbal communication, sports or human-robot interaction. This would be a future work direction.

**Acknowledgments**

The research leading to these results is funded by the German Federal Ministry for Economic Affairs and Energy within the project "KI Delta Learning" (project number: 19A19013A). The authors would like to thank the consortium for the successful cooperation.

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
