# OpenReview forum: "Anomaly Detection in Multi-Agent Trajectories for Automated Driving"
_robot-learning.org/CoRL/2021/Conference — CoRL2021 Poster_

### Official Review · Reviewer_tMsu · 2021-07-17

**Originality:** Good
**Technical Quality:** Good
**Clarity Of Presentation:** Good
**Impact:** 3

**Recommendation:**

Weak Accept: I recommend accepting the paper, but will not argue for my recommendation if the majority of other reviewers have a different opinion.

**Summary:**

This paper makes two contributions: (1) it proposes a spatio-temporal graph auto-encoder for learning nominal driving behaviors, after which kernel density estimation is performed on the latent state to quantify unlikely (i.e., “anomalous”) latent states, and (2) develops a dataset of nominal and anomalous driving behavior with associated human labels.

**Issues:**

- Clarify how the proposed approach differs from state-of-the-art trajectory predictors (cited in the detailed comments).
- Clarify why the measure of anomaly studied in this work is a sufficient metric


**Reviewer Expertise:**

Good: General knowledge of the area

**Strengths And Weaknesses:**

Strengths:
- The contribution of a novel dataset of two-agents driving on a highway with associated human anomaly labels is interesting and relevant to the autonomous vehicle community.
- The problem domain of anomaly detection in multi-agent trajectories is highly relevant for modern autonomous vehicle systems.

Weaknesses:
- The core contribution in this work is “the ability to simultaneously learn multiple trajectories for a dynamic number of agents” and then “perform kernel density estimation on the latent representation of the [proposed approach].” However, this has been extensively studied in the behavior prediction domain, it is not immediately clear why the proposed approach is a better paradigm since there is no discussion or comparison to state-of-the-art approaches which meet both these criteria. Further explanation is described in the detailed comments.

Detailed comments:

Section 1:
- “...e.g., recognizing a ghost driver…”, please clarify / define what a ghost driver is, as this is not standard terminology.
- “the main innovation of STGAE is the ability to simultaneously learn multiple trajectories for a dynamic number of agents'' and  then “perform kernel density estimation on the latent representation of the [proposed approach].” This idea has been studied in various ways by industry behavior prediction teams [1, 2] and academia [3]. In the proposed method, kernel density estimation is performed on the latent representation, and the density estimates are then utilized to detect anomalies. In the aforementioned cited approaches, the distribution over the latent state is often directly estimated, and then coupled with the trajectory forecasting method as a means to quantify likely and unlikely behaviors. For this reason, the novelty of the proposed approach should be discussed / compared to the aforementioned methods which capture similar distributions over latent states as well as produce trajectory predictions for a dynamic number of agents.
[1] Chai, Yuning, et al. "Multipath: Multiple probabilistic anchor trajectory hypotheses for behavior prediction." arXiv preprint arXiv:1910.05449 (2019).
[2] Tolstaya, Ekaterina, et al. "Identifying driver interactions via conditional behavior prediction." arXiv preprint arXiv:2104.09959 (2021).
[3] Ivanovic, Boris, et al. "Multimodal deep generative models for trajectory prediction: A conditional variational autoencoder approach." IEEE Robotics and Automation Letters 6.2 (2020): 295-302.
- The paper does not upfront define what is meant by “anomaly” in this work. Anomalies can take many different forms in multi-agent trajectory forecasting (e.g., is an agent’s true latent state in the hypothesis space of the model? Is the distribution over states well-calibrated? Assuming the probabilistic model is well-calibrated, is an agent performing a sufficiently unlikely maneuver deemed “anomalous”? What constitutes “sufficiently unlikely”?). It is only in Section 3.2 where the idea of anomalies being samples taken from low-density regions of the latent feature space is described. Being upfront about what anomaly means in this work would strengthen and clarify the narrative. Additionally, a discussion on why this notion of anomaly is valuable compared to alternative notions would also be valuable.

Section 3:
- Conditional variational autoencoder approaches (CVAE) seem well-suited for the proposed approach, as they both produce a conditional probability distribution given a dataset of driving trajectories but also project the inputs into a lower latent space and a distribution over the latent variable. It would be valuable to discuss why the proposed approach -- which performs kernel density estimation on the latent representation of the STGAE -- is different / more beneficial for autonomous vehicle trajectory modelling compared to the CVAE approaches.
- “To identify if a frame is normal or abnormal we compute the frame anomaly score… max-pooling avoids missing anomalies compared to the mean, which is not justifiable in automated driving.” While it is true that anomalies compared to the mean are not justifiable, looking at the mean when detecting anomalies may not be sufficient, as anomalies are often tail-events. This section would be strengthened with a further discussion on why looking at the mean during anomaly detection is a good measure of an anomaly. For example, why not detect the most-anomalous agent and latent state?
- At the end of this section, it would be valuable to describe what should be done (from an AV design and/or decision-making perspective) once an anomaly is detected. This would improve the impact of this work for the broader AV community that would have to integrate an anomaly detection framework into the vehicle perception/prediction/planning/control stack.

Section 4.3:
- Please define AUROC, FPR, and TPR prior to using the abbreviation.


**Summary Of Recommendation:**

I recommended weak rejection because although the domain of anomaly detection for autonomous driving prediction is highly relevant, the proposed method is not clearly described and evaluated compared to state-of-the-art trajectory forecasting methods which both predict multi-agent trajectories but also learn distributions over latent states. This paper would have been stronger with a focus on the design and collection of a novel anomaly-centric dataset, since this is something thoroughly lacking in the autonomous vehicle safety community.

====================
After the rebuttal phase, I have increased my recommendation to "weak accept".

---

> ### Author Response · Authors · 2021-08-27
> **Response to Reviewer tMsu**
>
> We thank the reviewer for the thorough review and address all comments and suggestions in the following. All changes in the **revised** paper and the supplementary material are marked in *italics* and blue.
>
> ***Q1: Please clarify/define what a ghost driver is, as this is not standard terminology.***
>
> For clarity, we changed "ghost driver" to "wrong-way driver" throughout the paper.
>
> ***Q2: The novelty of the proposed approach should be discussed/compared to the behavior prediction methods [1, 2, 3] which capture similar distributions over latent states as well as produce trajectory predictions for a dynamic number of agents.***
>
> The task of trajectory prediction deals with future outputs. Our work deals only with the past outputs up to the current time step. Moreover, anomaly detection of agent trajectories is an unsupervised one-class classification approach (no labels involved). Trajectory prediction is thus related, but it addresses a different problem. The provided references ([1-3]) deal with trajectory prediction only. Transforming them to anomaly detection is not straightforward and also out of the context of our work.
>
> ***Q3: It would be valuable to discuss why the proposed approach -- which performs kernel density estimation on the latent representation of the STGAE -- is different / more beneficial for autonomous vehicle trajectory modeling compared to the CVAE approaches.***
>
> The Variational Auto-encoder (VAE) learns a representation of a fixed-size input in the spatial domain. Unlike, our model learns a latent representation for a varying amount of agents individually in the spatio-temporal domain. VAEs have been extended to the temporal domain for anomaly detection on sensor signals by combining with a Long Short-Term Memory (LSTM) network [1]. However, it is assumed a fixed number of input streams, i.e. sensor signals, instead of a varying number of trajectories (our approach), i.e. agents on a highway. In an autonomous driving setting, the number of agents can vary a lot from a single agent on a lonely highway to many interacting agents in a traffic jam. The method we propose can scale with the number of agents and is therefore of higher practical impact for automated driving scenarios. We add the corresponding discussion to the related work section. In the supplementary material A.3, we show an ablation study on how the anomaly detection scales as the number of agents on the highway increases.
>
> *[1] Park, Daehyung, Yuuna Hoshi, and Charles C. Kemp. "A multimodal anomaly detector for robot-assisted feeding using an lstm-based variational autoencoder." IEEE Robotics and Automation Letters 3.3 (2018).*
>
> ***Q4: Further discussion on why looking at the mean during anomaly detection is a good measure of an anomaly. For example, why not detect the most-anomalous agent and latent state?***
>
> For the evaluation of our approach, we adopt the protocol of Morais et al. [1]. They compute an anomaly score as the maximum of all skeleton instances appearing in a frame. Similarly, we compute the anomaly score of one frame as the maximum over the anomaly scores of all agents present in the frame, see Eq. 8. This results in the most anomalous agent producing the final frame anomaly score. The anomaly score of one agent is computed by the KDE given the latent state of the agent.
>
> *[1] Morais, Romero, et al. "Learning regularity in skeleton trajectories for anomaly detection in videos." Proceedings of the IEEE/CVF Conference on Computer Vision and Pattern Recognition. 2019.*
>
> ***Q5: What should be done (from an AV design and/or decision-making perspective) once an anomaly is detected?***
>
> Once an anomaly is detected, the vehicle should react or pass the control to the human. We added discussion for the post-detection phase in the experimental section.
>
> ***Q6: Please define AUROC, FPR, and TPR prior to using the abbreviation.***
>
> We define the metric abbreviations in section 4.3 and accentuate the corresponding sections.

---

> > ### Comment · Reviewer_tMsu · 2021-09-03
> > **Re: author comments**
> >
> > I would like to thank the authors for their detailed responses and updates to the manuscript. I also appreciated the ablation study on how the anomaly detection scales as the number of agents on the highway increases.
> >
> > Several of my concerns have been addressed and so I am increasing my score to weak accept.

---

### Official Review · Reviewer_McLa · 2021-07-23

**Originality:** Very Good
**Technical Quality:** Very Good
**Clarity Of Presentation:** Very Good
**Impact:** 3

**Recommendation:**

Weak Accept: I recommend accepting the paper, but will not argue for my recommendation if the majority of other reviewers have a different opinion.

**Summary:**

The authors develop a system for anomaly detection in driver interactions using a novel spatio-temporal graph neural network architecture combined with a Kernel Density Estimator.


**Issues:**

In Table 2, it would be helpful to indicate which are the single agent anomalies, and which describe interaction between agents.

It would be good to discuss the possible uses & applications of anomaly detection in autonomous driving to further motivate the work.

**Reviewer Expertise:**

Very good: Comprehensive knowledge of the area

**Strengths And Weaknesses:**

The definition of a spatio-temporal graph and its use in a GNN architecture is a very interesting contribution of the work.
The authors provide relevant baselines for comparison, and perform several ablation studies.
The results clearly demonstrate that considering multi-agent interactions in anomaly detection is essential. The characterization of the performance of the system over various segment lengths is also important.

One shortcoming of the approach is that the kernel density estimator must use all of the training samples. Many of the most popular autonomous driving datasets have millions of examples, and it is unclear how the KDE methodology would scale to larger datasets.
I would like to see a discussion of how this approach could scale beyond pairs of agents, since the most challenging and interactive scenarios occur in dense urban traffic.

The current pipeline for anomaly detection could be used for data mining in a larger dataset to find interesting long-tail cases for training or testing components of autonomous cars. I think this system could also be useful for anomaly prediction, or early detection of situations that may become anomalous in the near future. The use of the max-pooling operation in Eq. 8 may allow for this, especially considering the consistent performance across segment lengths in Fig. 3. If this approach shows promise for this task, then the anomaly detection system could be used on-board an autonomous vehicle to detect when the car would need to revert to a human fall-back driver.


**Summary Of Recommendation:**

The paper describes a novel architecture for processing interactive trajectories and detecting anomalies, but the applicability of it to large modern datasets is limited by the use of KDE.

---

> ### Author Response · Authors · 2021-08-27
> **Response to Reviewer McLa**
>
> Thank you for the detailed review and constructive comments. All changes in the **revised** paper and the supplementary material are marked in *italics* and blue.
>
> ***Q1: I would like to see a discussion of how this approach could scale beyond pairs of agents since the most challenging and interactive scenarios occur in dense urban traffic.***
>
> Thank you for the comment. We add an ablation study on the performance beyond pairs of agents to the supplementary A.3. To this end, we augment a subpart of the test set with two more agents. For visualizations of example sequences, we refer to Figure 4 in the supplementary material. The final dataset includes 11 abnormal and 11 normal sequences.
>
> From the new test set, we design three sets with a different amount of agents. This means, we ignore either both new agents, $N=2$, one of the new agents, $N=3$, or use all four agents, $N=4$. The table below summarizes the results. We compare our method with the reconstruction method *STGAE-biv*. Interestingly, the metrics of the reconstruction method vary intensively with the number of agents. It looks different for our method. Even though we see a small performance decrease with an increasing number of agents, our method is more stable and can reliably detect anomalies on highways with higher traffic density.
>
> | Test Agents 	| Model     	| AUROC (⭡)	| AUPR-Abnormal (⭡)	| AUPR-Normal (⭡)	| FPR-95%-TPR (⭣)	|
> |----------	|-----------	|-------	|---------------	|-------------	|-------------	|
> | $N=2$    	| STGAE-biv 	| 69.08 	| 39.28         	| 90.07       	| 92.19       	|
> | $N=2$    	| Ours      	| 92.34 	| 66.75         	| 98.48       	| 35.20       	|
> | $N=3$    	| STGAE-biv 	| 78.20 	| 42.52         	| 94.73       	| 78.23       	|
> | $N=3$    	| Ours      	| 91.26 	| 63.57         	| 98.36       	| 35.38       	|
> | $N=4$    	| STGAE-biv 	| 52.42 	| 17.95         	| 88.03       	| 88.19       	|
> | $N=4$    	| Ours      	| 89.41 	| 60.52         	| 97.87       	| 42.55       	|
>
> ***Q2: How would the KDE method scale to larger datasets?***
>
> The KDE scales with time complexity $\mathcal{O}(|\mathbf{Z}|)$ as the number of training feature vectors $|\mathbf{Z}|$ increase. For large datasets we find random sub-sampling an efficient technique to keep the real-time capability. Simultaneously, the performance remains stable over a large range of sampling rates. We add the discussion on the scalability w.r.t. the training dataset size to the supplementary A.4.
>
> ***Q3: In Table 2, it would be helpful to indicate which are the single-agent anomalies, and which describe interaction between agents.***
>
> As suggested, we indicate the three single-agent and eight interactive-agent types of anomalies in a separate column in Table 2.
>
> ***Q4: It would be good to discuss the possible uses & applications of anomaly detection in autonomous driving.***
>
> An anomaly detection module directly supports the decision-making process. Once an anomaly is detected, the automated vehicle should react or pass the control to the human driver. We added discussion in the experimental section.

---

### Official Review · Reviewer_BxGJ · 2021-08-01

**Originality:** Good
**Technical Quality:** Good
**Clarity Of Presentation:** Very Good
**Impact:** 2

**Recommendation:**

Weak Accept: I recommend accepting the paper, but will not argue for my recommendation if the majority of other reviewers have a different opinion.

**Summary:**

This paper proposes a method for anomaly detection for the autonomous driving domain, by learning a representation for joint trajectories of normal driving behaviors, then performing density estimation on the latent representation of the joint trajectory auto-encoder.  At test time, anomalies are detected by test trajectories occurring in low density regions of the learned latent space.  In particular, the proposed method consists of first training a spatio-temporal graph CNN auto-encoder (STGAE) to both represent individual “normal” trajectories in a latent feature space (autoencoder) and model agent (car) interactions over time by computing pairwise agent influences (spatio-temporal graph).  Secondly, it uses kernel density estimation on the latent feature representation of the STGAE to approximate the distribution of the normal trajectories.  The proposed method is evaluated against single-agent and multi-agent baselines, on standard metrics for anomaly detection.  The paper also contributes a new simulated dataset for multi-agent driving trajectories, with both normal and abnormal driving maneuvers.

**Issues:**

Listed above as suggestions for improvement.

**Reviewer Expertise:**

Good: General knowledge of the area

**Strengths And Weaknesses:**

Identified strengths of paper:
+  Addresses an interesting and under-explored problem in multi-robot systems, anomaly detection given joint continuous trajectories; thus it explores the detection/recognition of anomalous behaviors for multi-robot systems, intended for the autonomous driving domain.  I also wonder how well this approach would generalize to other MA problem domains with continuous action spaces?  E.g. multi-agent nonverbal communication or coordination in a shared environment.  If the authors have insights about this, some discussion of it at the end of the paper would be interesting and potentially useful.
+  Provides extensive empirical analysis, which is important for extracting meaningful insights.  Good that the work evaluates methods using several metrics from the Anomaly Detection literature; provides a stronger basis for hypothesis testing.  Qualitative analysis from Table 2 is useful for providing additional insight into aggregate results and also seems like good evidence for differentiating between the value of single-agent versus multi-agent methods on this problem.
+  Contributes a simulated driving dataset to the MA community, with 146 scenes and 11 different types of driving anomalies, which can be a really useful tool for the community in a problem domain of interest to many.


I would suggest that the authors improve their submission, including the following:
-  For all results reported: Would be extremely useful (and provide more compelling evidence) if some measure of statistical dispersion was included (e.g. standard deviation or standard error), so as to assess statistical significance when comparing the different methods.  Currently, this is not shown at all, only mean data.  So it is difficult to assess the reliability of the findings.  Also, how many seeds/runs are these curves averaged over?
-  The paper briefly explains each of the evaluation metrics used, which is really helpful as context.  However, even with that, it's not entirely obvious to me how to interpret each of the result columns in Table 1.  I understand for certain columns, lower/higher is better, but what do the differences in performance imply?  How much of a difference is significant?  Overall, it would be helpful to provide additional explanation/intuition as to how to interpret the findings, particularly for readers not from the Anomaly Detection community.  Also, still unclear as to *why* all baselines are not shown in Figure 3.
-  From a qualitative analysis perspective, it could be insightful to show an example of where the same trajectory for agent i is normal in the single-agent case but can be anomalous in the multi-agent case (dependent upon the behavior by another agent j).  Would also serve as additional compelling motivation for modeling of joint continuous trajectories, which I’m wondering how much more computationally expensive it is than modeling only individual trajectories? Some discussion about computational complexity would be useful.
-  Minor Clarification:  In Equation 5, remember to define p^i_t


**Summary Of Recommendation:**

This paper makes a novel methodological contribution to the space of multi-agent robotic or dynamical systems, by applying spatio-temporal GNNs for anomaly detection.  While the work is intended for a specific application domain (autonomous driving), which serves to limit the generality of its findings, the method considers learning in continuous joint trajectory spaces, which contributes to an under-explored part of the multi-agent reinforcement learning (MARL) problem space.  The application domain itself is of high relevance and interest to the MARL and Robotics communities, particularly as the former is moving toward developing methods for more complex domains.  And in the application domain of interest, the paper is able to show that the proposed method benefits from modeling the interactive (multi-agent) nature of the driving problem, not just considering the single-agent problem.  It also provides insights on the types of driving anomalies where the method performs well.  Finally, it contributes a simulated driving dataset, which is a nice contribution to the community.

---

> ### Author Response · Authors · 2021-08-27
> **Response to Reviewer BxGJ**
>
> Thank you for the detailed review, questions, and suggestions. Below, we address them and incorporated all the feedback. All changes in the **revised** paper and the supplementary material are marked in *italics* and blue.
>
> ***Q1:  I wonder how well this approach would generalize to other MA problem domains with continuous action spaces?***
>
> The proposed model can be easily extended to different types of input features. For example, it can be used to detect anomalies in human skeleton motion in nonverbal communication or sports, the spoken words in inter-human communication, or the motion in multi-robot or human-robot interaction. We added a discussion to the conclusion section.
>
> ***Q2: It would be extremely useful if some measure of statistical dispersion was included.***
>
> In Table 1, we added the statistical dispersion to the baseline comparison. Each model is trained three times with the same hyperparameters but different network initialization. We report the mean and standard deviation. In total, the one-class methods show lower variance over multiple runs for all metrics. The AUPR-Abnormal metric shows a higher variance compared to AUPR-Normal. This is due to the class imbalance towards the normal class where 85% of the test frames are normal, while 15% are abnormal. Nevertheless, our approach shows the lowest variance of all multi-agent models in the AUPR-Abnormal metric. Note that CVM and LTI are deterministic with zero standard deviation.
>
> ***Q3: What do the differences in performance in Table 1 imply?***
>
> Table 1 evaluates the recognition performance on the four standard metrics in anomaly detection. First, AUROC summarizes the receiver-operating characteristic, which opposes TPR to FPR for different anomaly score thresholds as shown in Figure 2. In a practical application, one would choose a particular threshold on the ROC curve to detect anomalies. Similar, FPR-95%-TPR measures the FPR in an operating point with TPR equal to 95%. Lower values are better as they result in fewer false alarms. Both AUPR-Abnormal and AUPR-Normal summarize the area under the precision-recall curve. To evaluate statistical significance, we average the metrics over three runs, see Q2. Our approach has the advantage over the baselines as it remains superior to the other models. We added the short discussion in the experiment section.
>
> ***Q4: Also, still unclear as to why all baselines are not shown in Figure 3.***
>
> Thank you for the recommendation. In the revised version we compare all baselines with our model over various segment lengths. In general, the reconstruction methods drop in performance for longer segments. Both one-class classification methods, *STGAE-biv+OC-SVM* and our approach, provide stable results over a long range of input sequence length, while our approach performs best overall.
>
> ***Q5: Some discussion about computational complexity would be useful.***
>
> During inference, the trajectory encoding with the STGAE is independent of the number of training samples. The time complexity of the KDE scales with $\mathcal{O}(|\mathbf{Z}|)$ with the number of training feature vectors $|\mathbf{Z}|$. We added a discussion to the supplementary material A.4.
>
> ***Q6: Minor Clarification: In Equation 5, remember to define p^i_t***
>
> We assume the coordinates $(x_t^i, y_t^i)$ follow the bivariate Gaussian distribution $p_t^i \sim \mathcal{N}(\mu_{t}^i ,\sigma_{t}^{i}, \rho_t^i)$ with mean $\mu_{t}^i$, standard deviation $\sigma_{t}^{i}$, and a correlation factor $\rho_t^i$. See line 140 in the paper.

---

> > ### Comment · Reviewer_BxGJ · 2021-09-03
> > **Response to Author Comments**
> >
> > I thank the authors for their detailed response to my questions and concerns, as well as the updates made accordingly to the paper.
> >
> > After reading the author response along with other reviewers' concerns, I still believe this work to make an interesting contribution.  Thus my score and recommendation remain unchanged.

---

### Meta-Review · Area_Chair_DJ8Z · 2021-08-15

**Recommendation:** Accept (Poster)
**Confidence:** 4

**Metareview:**

The paper introduces a novel anomaly detection mechanism for the multi-agent, autonomous driving domain. All reviewers agree that the paper addresses an important and timely problem for autonomous vehicles. The reviewers expressed concerns about the performed
evaluation that should be addressed in the revised version:
- Clarify the results and the inconsistencies represented in the tables and Fig. 3. It seems that the statistical spread in the performance of the proposed method and other baselines is missing. A more detailed discussion of the presented results will strengthen the paper significantly (see reviewers' suggestions).
- Clarify how the approach scales in terms of multi-agent interactions (as opposed to a pairwise interaction) and as the number of data samples increase.
- Discuss how the proposed approach differs from some of the related multi-agent trajectory prediction efforts pointed out by the reviewers.
- Address the suggested language improvements.

===== Post rebuttal =====

The authors have sufficiently addressed reviewers' concerns; specifically, ablation studies with multi-agent interactions were particularly interesting. For the final version, I recommend the authors to add some of these studies in the main paper. I would also suggest doing a more thorough statistical analysis (3 trials are not that significant to compute the mean/variance of the performance).

---

> ### Author Response · Authors · 2021-08-27
> **Response to the Area Chair DJ8Z**
>
> We thank the AC and the reviewers for giving constructive feedback and address all comments in the following and in the individual responses. All changes in the **revised** paper and the supplementary material are marked in *italics* and blue.
>
> ***AC1. Clarify the results and the inconsistencies represented in the tables and Fig. 3. It seems that the statistical spread in the performance of the proposed method and other baselines is missing.***
>
> Following the suggestion of the reviewers, we average the results in Table 1 over three runs and show the mean and standard deviation. The proposed approach reaches high scores compared to the baselines with lower variance over multiple runs. As suggested by the reviewers, we compare the performance of all methods on the input segment length in Fig. 3.
>
> ***AC2. Clarify how the approach scales in terms of multi-agent interactions (as opposed to a pairwise interaction) and as the number of data samples increase.***
>
> We provide an experimental setting with two, three, and four agents in the supplementary material A.3. It shows evidence that the proposed model is not disturbed by additional interactions and can still provide high recognition rates in more complex scenarios.
> In addition, we study the inference time and the performance of the model on different amounts of data samples in supplementary A.4.
>
> ***AC3. Discuss how the proposed approach differs from some of the related multi-agent trajectory prediction efforts pointed out by the reviewers.***
>
> Anomaly detection is an unsupervised one-class classification approach, while trajectory prediction deals with the prediction of future agent states. Both are trained on multi-agent trajectories, but only anomaly detection deals with the past agent states. We replied to the reviewer [tMsu] in more detail.
>
> ***AC4. Address the suggested language improvements.***
>
> We correct equation (5) [BxGJ] and define the abbreviations of the metrics [tMsu].

---

### Decision · Program_Chairs · 2021-09-13

**Decision:**

Accept (Poster)

**Comment:**

The paper introduces a novel anomaly detection mechanism for the multi-agent, autonomous driving domain. All reviewers agree that the paper addresses an important and timely problem for autonomous vehicles. The reviewers expressed concerns about the performed
evaluation that should be addressed in the revised version:
- Clarify the results and the inconsistencies represented in the tables and Fig. 3. It seems that the statistical spread in the performance of the proposed method and other baselines is missing. A more detailed discussion of the presented results will strengthen the paper significantly (see reviewers' suggestions).
- Clarify how the approach scales in terms of multi-agent interactions (as opposed to a pairwise interaction) and as the number of data samples increase.
- Discuss how the proposed approach differs from some of the related multi-agent trajectory prediction efforts pointed out by the reviewers.
- Address the suggested language improvements.

===== Post rebuttal =====

The authors have sufficiently addressed reviewers' concerns; specifically, ablation studies with multi-agent interactions were particularly interesting. For the final version, I recommend the authors to add some of these studies in the main paper. I would also suggest doing a more thorough statistical analysis (3 trials are not that significant to compute the mean/variance of the performance).